# Long-Term Effect of Tamarisk Plantation on Soil Physical Properties and Soil Salt Distribution in Coastal Saline Land

**Jingsong Li** [1,2], **Ce Yang** [1,2], **Tabassum Hussain** [3], **Xiaohui Feng** [1,2], **Xiaojing Liu** [1,2] and **Kai Guo** [1,2,*]

1   Key Laboratory of Agricultural Water Resources,
    CAS Engineering Laboratory for Efficient Utilization of Saline Resources,
    Center for Agricultural Resources Research, Institute of Genetics and Developmental Biology,
    Chinese Academy of Sciences, Shijiazhuang 050021, China
2   University of Chinese Academy of Sciences, Beijing 100049, China
3   Dr. M. Ajmal Khan Institute of Sustainable Halophyte Utilization, University of Karachi,
    Karachi 75270, Pakistan
*   Correspondence: guokai@sjziam.ac.cn

**Abstract:** Ecological restoration of coastal land by planting salt-tolerant plants has been widely used to construct vegetation. The aim of this study was to evaluate the changes induced by tamarisk (*Tamarix chinensis*) shrub on coastal soil physical quality and as well as the corresponding impact on salt distribution in the soil. A field study was conducted on coastal saline land, North China, where tamarisk plantation was established 5-year-old (T-5yr) and 11-year-old (T-11yr), and compared with barren land as control (CK). Quantitative soil physical properties, soil physical quality index, soil salt distribution, and salt leaching were examined. The results indicated that planting tamarisk improved the coastal soil properties at higher degree in topsoil than in deep soil layers. Tamarisk plantation significantly increased soil organic carbon content and pH. It also enhanced the formation of soil large aggregates and porosity; however, reduction soil bulk density and salt content in topsoil were recorded. Soil physical quality index was positively correlated with root weight density of tamarisk, and soil of T-11 yr plantation exhibited the highest soil physical quality index, with promoted soil physical functions of supporting root growth and the resistance to soil degradation. In addition, tamarisk induced soil physical changes which enhanced the salt-leaching in rainy season and contributed to the homogeneous salt distribution in soil profile. Consequently, the ecological benefits of tamarisk vegetation turned coastal saline land into a fertile land by plant–soil interaction and the soil structure improvement, therefore, it prevented the natural soil accumulation by accelerating the salt leaching after tamarisk was restored. This study provides some insights into the mechanism of tamarisk on coastal soil restoration and its regulation of soil salt distribution.

**Keywords:** coastal saline land; tamarisk; ecological restoration; soil salinity; soil physical functions

## 1. Introduction

Soil salinization is an environmental problem worldwide, and it has become prominent because of the rapid global climate change [1]. In coastal regions, seawater intrusion and capillary water rising from the shallow saline groundwater caused the increase of soil salinity, thereby limiting the survival of most plants [2–4]. The lack of protective vegetation in coastal land resulted in the poor ecological functions of preventing windstorms, fastening sand, resisting the harm of tsunami and storm surge, and beautifying the living environment [5,6]. Thus, it was of great significance to construct vegetation to maintain ecological security of coastal zones. Recently, utilizing natural salt-tolerant plants was considered a cost effective and sustainable solution to reconstruct vegetation for ecological restoration in coastal saline land [7].

Tamarisk (*Tamarix chinensis*) is one of the few native salt-excreting shrubs, widely distributed in saline soil from semi-arid inland to coastal region [8]. With the evolutionary

trait of excreting ions in leaves, it evolved strong salt-tolerant capabilities and can survive up to 30 g/kg salt content in the soil [9,10]. Studies of Xia et al. [11] and Newete et al. [12] have revealed the adaptive strategies of tamarisk to salinity and suggested the potential availability of this species on ecological restoration in saline-alkali lands. However, the impacts of tamarisk on soil environment and the corresponding ecological effects are still needed to be explored.

Some studies indicated that tamarisk vegetation resulted in an increase of soil nutrients under plant canopies and positively influenced on soil properties [13–19]. While most of these studies focused on soil chemical and biological improvement, the effect of tamarisk on soil physical changes was seldom reported. Paradoxically, due to the capacity of tamarisk to secreting absorbed salts, the increase of salt content and pH in the soil after tamarisk restoration was also observed, which suggested the potential risks of soil salinization and negative ecological effects [20,21]. Therefore, exploring the evolution of soil physical properties, soil salt content changes, and their relationships in a long-term scale can provide guidance for the restoration, utilization, and prediction of soil salinization in coastal region due to planting tamarisk.

It was acknowledged that plant–soil feedback was a complex ecological issue that is tightly associated with the environment [22,23]. In salt-affected land, there were close relationships among vegetation, water transport, and salts distribution across the groundwater-vadose, zone–aquifer, and soil layer [24,25]. Thus, we hypothesized that tamarisk induced soil physical changes played an important role in regulating soil salt distribution. In addition, how the spatiotemporal distributions of soil water and salts develop following vegetation construction is an important issue to address the ecological restoration in coastal land [26].

In the Chinese "Southern Mangrove and Northern Tamarisk" ecological project, tamarisk was widely planted at coastal lands and has showed advantages in vegetation construction over the past decade [27–29]. In this context, a field study was conducted to evaluate the effect of tamarisk vegetation (5-year-old and 11-year-old plantations) on soil physical properties, soil physical quality index, and soil salt distribution. The aim of this study was to explore the regulation of coastal soil changes following tamarisk restoration in a broad time-scale.

## 2. Material and Methods

### 2.1. Site Description and Tamarisk Plantation

This study was carried out at the coastal land near Bohai Bay, located in Haixing County, Hebei province, North China (117°57′17″–117°58′31″ E, 38°16′83″–38°17′59″ N) (Figure 1a). The study area has a semi-humid continental monsoon characterized by a short rainy summer, and long dry spring, autumn, and winter. The annual average temperature was 15.1 °C and annual average precipitation was 570 mm, and approximately 75% of the annual precipitation occurs during July to September [28,30]. The annual average evapotranspiration was about 1240 mm and higher evapotranspiration rates than precipitation resulted in soil salt accumulation and the increase of topsoil salinity. The average elevation for study site is 2.8 m and the field site sits atop a shallow groundwater table that maintains a high salinity range (10–40 g L$^{-1}$) [31,32]. The study site was 24.8 km away from the coastline and it was a typical dry land affected by soil salinization.

Before re-vegetation, the study site was abandoned land with high soil salt content (about 30 g kg$^{-1}$) and there was little artificial disturbance for decades [31]. The soil in this region has a silty clay loam texture and classified as inceptisols, according to NRCS soil classification system [33]. The vegetation community was mostly composed of halophytes, including *Suaeda salsa*, *Suaeda glauca*, *Aeluropus sinensis*, *Phragmites australis*, *Ixeris chinensis*, etc.

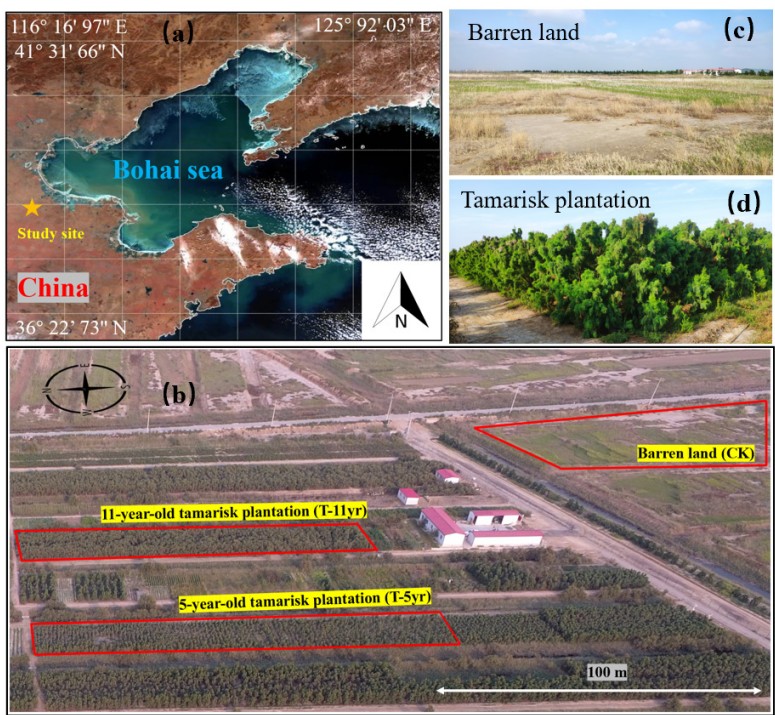

**Figure 1.** Geographical location of study site (**a**), sample sites map (**b**), photos of barren land (**c**), and tamarisk plantation (**d**).

Tamarisk (*Tamarix chinensis*) plants were planted in the origin land of study site respectively in 2009 (11-year-old plantation; T-11yr) and 2015 (5-year-old plantation; T-5yr) through branch cuttings cultivation (Figure 1c). After being stored in sand, tamarisk cuttings were planted in the density of $2.91 \times 10^4$ plant/ha and then grew naturally. The area of 5-year-old and 11-year-old tamarisk plantation land was about 0.22 and 0.19 ha, respectively. Furthermore, barren land (about 0.32 ha) in the same plot was selected as the control (CK) which was used to reflect the soil properties and performance without the impacts of tamarisk (Figure 1c).

### 2.2. Sampling and Measurement

In April 2020, four individual tamarisk plants were selected randomly in each plantation for measuring the tree height, diameter, and biomass of leaf and stem. Dig method was used for sampling plant root as described in [34]. After being cleaned and oven-dried, root samples were weighed to calculate the root weight density. The plant population characteristics were listed in Table 1.

**Table 1.** The plant population characteristics for 5-year-old (T-5yr), and 11-year-old tamarisk plantation land (T-11yr).

| | Height (m) | Diameter (cm) | Above-Ground Biomass (kg/Plant) | | Specific Root Weight (kg/m³) | | | Litter Biomass (kg/m²) |
| --- | --- | --- | --- | --- | --- | --- | --- | --- |
| | | | Leaf | Stem | 0–20 cm | 20–40 cm | 40–60 cm | |
| T-5yr | 1.67 ± 0.08 | 3.54 ± 0.40 | 0.70 ± 0.12 | 0.90 ± 0.21 | 1.79 ± 0.88 | 2.19 ± 0.81 | 1.32 ± 0.67 | 0.94 ± 0.21 |
| T-11yr | 2.43 ± 0.20 | 5.06 ± 0.75 | 0.98 ± 0.16 | 2.42 ± 0.60 | 3.46 ± 0.97 | 1.76 ± 0.65 | 1.04 ± 0.36 | 1.31 ± 0.27 |

Values represent means ± S.E ($n = 4$).

Soil samples were collected in 0–20, 20–40, and 40–60 cm layers from barren land (CK), 5-year-old (T-5yr), and 11-year-old (T-11yr) tamarisk experimental plots. In each layer, 100 cm³ steel rings were used to measure the soil physical properties, including soil bulk density (BD), macro—(MA), meso—(ME), micro-porosity (MI), soil water storage capacity (SWSC), and soil aeration capacity (SAC). Undisturbed soil block was used for soil water-stable aggregates analysis, and transferred to a plastic tray to perform the visual

evaluation using the visual evaluation of soil structure (VESS) method, as described by Guimarães et al. [35]. We applied progressive force by first attempting to squash soil sample between the fingers, then in one hand and then with both hands. It was compared with a visual key attribute score of the soil structural layers in the slice. Size, strength, porosity, roots, and color are the main criteria used to define soil score and the score ranges from 1 (good) to 5 (poor soil structure); disturbed soil samples were used to measure soil organic carbon content (SOC), soil water content (SWC), soil electrical conductivity (EC), soil pH, and soluble ions content (sum of $Na^+$ and $K^+$, $Ca^{2+}$, $Mg^{2+}$; $Cl^-$, $HCO_3^-$, and $SO_4^{2-}$). Four replicates were used for each sampling.

During the rainy season (From mid-June to mid-October), there was about 430 mm precipitation in 2020 (the data were collected from a weather station). Thus, to examine the variation of soil salt distribution after an intensive rainfall event, we collected the soil samples on 15-June and 15-October to measure the soil salt content, as 20 cm intervals to 100 cm depth from barren land (CK), 5-year-old (T-5yr) and 11-year-old (T-11yr) tamarisk plantation lands, respectively.

In the laboratory, the soil cores samples covered with filter paper were soaked in fresh water for 24 h to water saturation. Then, soil cores were weighed under matric suction pressures of 30, 100, and 150 KPa to calculate the volumetric soil water content [34]. Finally, we measured the weight of the oven-dried soil to calculate the soil bulk density (BD) and soil total porosity (TP) with the following equation:

$$TP = (1 - BD/ds) * 100 \tag{1}$$

where soil particle density (ds) was assumed as 2.65 g/cm$^3$ [36].

Macro—(MA), meso—(ME), micro-porosity (MI), soil water storage capacity (SWSC), and soil aeration capacity (SAC) were calculated using the following equations:

$$MA = TP - \theta_{30KPa} \tag{2}$$

$$ME = \theta_{30KPa} - \theta_{100KPa} \tag{3}$$

$$MI = \theta_{100KPa} \tag{4}$$

$$SWSC = \theta_{FC}/TP \tag{5}$$

$$SAC = 1 - SWSC \tag{6}$$

where $\theta_{30KPa}$ and $\theta_{100KPa}$ are the volumetric soil water content at the matric suctions of 30 and 100 KPa, respectively. $\theta_{FC}$ was usually considered equal to $\theta_{30KPa}$ [37].

Soil organic carbon (SOC) was measured by potassium dichromate method [38]. Soil water-stale aggregate stability was determined by the standard wet method [39]. Soil mean weight diameter (MWD) was calculated using the following equation:

$$MWD = \sum X_i * W_i \tag{7}$$

where $X_i$ indicates the mean diameter of each size fraction and $W_i$ indicates the proportion of soil aggregate weights in the corresponding size.

The soil sample was oven-dried at 105 °C and weighed to determine the gravimetric soil water content (SWC). Soil electrical conductivity (EC) and pH were measured using 1:5 soil water extracts by conductivity meter (Horiba, B-173) and digital pH meter (Sartorius, PB-10), respectively. The main soil cations (sum of $Na^+$ and $K^+$, $Ca^{2+}$, $Mg^{2+}$) and anions ($Cl^-$, $HCO_3^-$, $SO_4^{2-}$) contents were measured by titration methods according to Guo and Liu [40]. Soil salts content (SAC) was calculated as the sum of soil main ion ($Cl^-$, $SO_4^{2-}$, $HCO_3^-$, $Na^+$, $K^+$, $Ca^{2+}$, and $Mg^{2+}$) contents.

## 2.3. Soil Physical Quality Index

Soil physical quality index (SPQI) was determined to evaluate the total effect of tamarisk on soil physical quality. According to Cavalcanti et al. [37] and Cherubin et al. [41], four critical soil physical functions were developed: f(i) soil capacity to support root growth; f(ii) balance between fluxes and storage capacity of water in the soil; f(iii) soil aeration capacity; and f(iv) soil resistance to degradation. Eight indicators were selected to compose the minimum dataset, including f(i): BD and VESS, f(ii): SWSC and ME, f(iii): SAC and MA, and f(iv): MWD and SOC. Each indicator was transformed into a unitless value ranging from 0 to 1 for calculation of the SPQI using the following equation:

$$SPQI = \sum f(scores)W \tag{8}$$

where SPQI is the soil physical quality index, f(scores) is the scores obtained for each function, and W is the weight of each function, as exemplified in Table 2.

**Table 2.** An example of the framework applied to develop the soil physical quality index (SPQI) for 5-year-old tamarisk plantation land (T-5yr) in 0–20 cm soil layer.

| Soil Physical Function | Weight (A) | Soil Indicators | Weight (B) | Value of Transformed Indicator (C) | Soil Indicator Score (B × C) | ∑(B × C) (D) | Soil Function Score (D × A) | SPQI ∑ (D × A) |
|---|---|---|---|---|---|---|---|---|
| f(i) | 0.25 | BD | 0.5 | 0.925 | 0.463 | 0.941 | 0.235 | |
| | | VESS | 0.5 | 0.957 | 0.478 | | | |
| f(ii) | 0.25 | SWSC | 0.5 | 0.838 | 0.419 | 0.782 | 0.200 | |
| | | ME | 0.5 | 0.726 | 0.363 | | | 0.652 |
| f(iii) | 0.25 | SAC | 0.5 | 0.718 | 0.359 | 0.430 | 0.107 | |
| | | MI | 0.5 | 0.141 | 0.071 | | | |
| f(iv) | 0.25 | MWD | 0.5 | 0.129 | 0.064 | 0.455 | 0.114 | |
| | | SOC | 0.5 | 0.782 | 0.391 | | | |

Note: f(i) is related to soil capacity to support root growth; f(ii) is related to the balance between fluxes and storage capacity of water in the soil; f(iii) is related to soil aeration capacity; f(iv) is related to soil resistance to degradation.

To explore the effect of the tamarisk roots on soil physical quality index (SPQI) changes, we collected the 0–20 cm depth SPQI in the lands of CK, T-5yr, and T-11yr, and then performed a linear fitting with the root weight density (RWD) of vegetation. Each site had three replicates.

## 2.4. Date Analysis

One-way analysis of variations (ANOVA) was conducted to investigate the differences in soil properties within different tamarisk reclamation years. Before this, the assumptions of normality and homogeneity of the variances of the residuals were tested by the Shapiro–Wilk and Levene tests. The mean comparisons were performed using Fisher's least significant difference (LSD) test with a probability defined at $p \leq 0.05$. The statistical procedures were performed using SPSS 16.0 software (SPSS Inc., Chicago, IL, USA).

## 3. Results

### 3.1. Soil Physical Properties

Tamarisk (T-5yr and T-11yr) significantly reduced the soil BD in 0–20 cm, where the following rank was observed T-11yr (1.32 g/cm³) < T-5yr (1.43 g/cm³) < CK (1.56 g/cm³). In deep soil layers (20–40 and 40–60 cm), tamarisk (T-5yr and T-11yr) had little influence on soil BD (Figure 2a). In CK, there was no variation of the visual evaluation of soil structure score (VESS) along the layers. The scores were about 4.5–4.75 for all soil layers. Compared with CK, tamarisk significantly reduced the value of VESS, and T-5yr and T-11yr showed similar values of VESS in the same layer (Figure 2b). Moreover, the VESS of T-11yr in 0–20 cm was 1.50, lower than that in the deeper soil layer (2.50 in 20–40 and 40–60 cm). Soil total porosity (TP) was increased from 0.42 m³/m³ (CK) to 0.45–0.49 m³/m³ (T-5yr and T-11yr) in the 0–20 cm layer (Figure 2c). Compared with CK, T-11yr significantly increased the macro-porosity (MA) in all the layers (Figure 2d). In 0–20 cm soil layer, tamarisk (T-5yr and

T-11yr) significantly reduced the meso-porosity (ME), while increasing the micro-porosity (MI) (Figure 2e,f). Overall, tamarisk not only increased the soil TP, but also changed the soil pore size, mainly the MA and ME distribution. Compared with CK, T-11yr significantly reduced the soil water storage capacity (SWSC) and increased soil aeration capacity (SAC) in all soil layers. In 0–20 cm, SWSC decreased from 0.80 (CK) to 0.66 after 11-years tamarisk restored (T-11yr), while there was no significant differnce between CK and T-5yr, for both SWSC and SAC in 0–20 cm soil layer. In 40–60 cm, the SAC of T-5yr was 0.17, significantly higher than that of CK (0.07) (Figure 2g,h).

Tamarisk (T-5yr and T-11yr) increased the soil organic carbon (SOC) and mean weight diameter (MWD) for all soil layers in relation to CK (Figure 3). In the 0–20 cm layer, the SOC and MWD in T-11yr were significantly higher than that in T-5yr. For soil water-stable aggregate size distribution (Figure 4), tamarisk (T-5yr and T-11yr) reduced <0.1 mm soil aggregate percentage and increased the part of >0.5 mm aggregate percentage in 0–20 cm layer. Moreover, the promotion effect of T-11yr on large soil aggregate (>0.5 mm) formation was higher than that of T-5yr in 0–20 and 40–60 cm layers. In 20–40 cm layer, tamarisk (T-5yr and T-11yr) significantly increased >2.0 mm aggregate percentage, while it had little effect on 1.0–2.0 mm soil aggregate percentage. In addition, the 0.5–1.0 mm soil aggregate percentage of T-11yr (23%) was significantly higher than that of CK (3%) and T-5yr (6%) in 20–40 cm layer.

### 3.2. Soil Physical Quality Index (SPQI)

In tamarisk plantations, the soil physical quality index (SPQI) in 0–20 and 20–40 cm was increased, while tamarisk had little influence on SPQI in 40–60 cm layer (Figure 5). Tamarisk (T-5yr and T-11yr) improved the soil physical functions, including supporting for root growth, water fluxes, and the resistance to soil degradation. In addition, from 5 years to 11 years following tamarisk restoration, the soil resistance to degradation was further improved, as in 0–20 cm (from 0.12 in T-5yr to 0.15 in T-11yr) and 40–60 cm layer (from 0.04 in T-5yr to 0.07 in T-11yr). Moreover, the SPQI increased with the increasing of root weight density (RWD) of tamarisk, and there was a linear correlation relationship between RWD and SPQI (Figure 6).

### 3.3. Soil Salt Distribution

Tamarisk (T-5yr and T-11yr) increased soil water content (SWC) in 0–20 cm layer and the SWC of T-11yr at 20–40 cm was 24.7%, significantly higher than that of T-5yr (23.0%) and CK (22.7%) (Figure 7a). Following tamarisk restoration, the soil EC was significantly reduced from 7.4 s/m (CK) to 2.3 s/m (T-5yr) and 2.5 s/m (T-11yr) in 0–20 cm (Figure 7b). Meanwhile, tamarisk (T-5yr and T-11yr) caused an increase of soil pH both in 0–20 and 20–40 cm layers (Figure 7c). There was no significant difference between soils beneath T-5yr and T-11yr in soil EC and pH.

As shown in Tables 3 and 4, $Cl^-$ and the sum of $Na^+$ and $K^+$ were the main salt ions composition in the study areas. The impact of tamarisk plantations on $Cl^-$ content was similar to that on the sum of $Na^+$ and $K^+$ content. $Cl^-$ and the sum of $Na^+$ and $K^+$ content of tamarisk plantations soil (T-5yr and T-11yr) were significantly lower than that of CK in 0–20 cm and 20–40 cm, while in 40–60 cm layer, however, tamarisk (T-5yr and T-11yr) caused an increase of $Cl^-$ and the sum of $Na^+$ and $K^+$ content in relation to CK. In 0–20 cm layer, tamarisk (T-5yr and T-11yr) had little impact on $Ca^{2+}$ content, but T-5yr significantly promoted it in 20–40 and 40–60 cm. Tamarisk (T-5yr and T-11yr) reduced the $Mg^{2+}$ content and increased the $HCO_3^-$ content in 0–20 cm. In addition, the $HCO_3^-$ content in T-5yr (0.38 g/kg) was significantly higher than that in T-11yr (0.35 g/kg) in 0–20 cm layer.

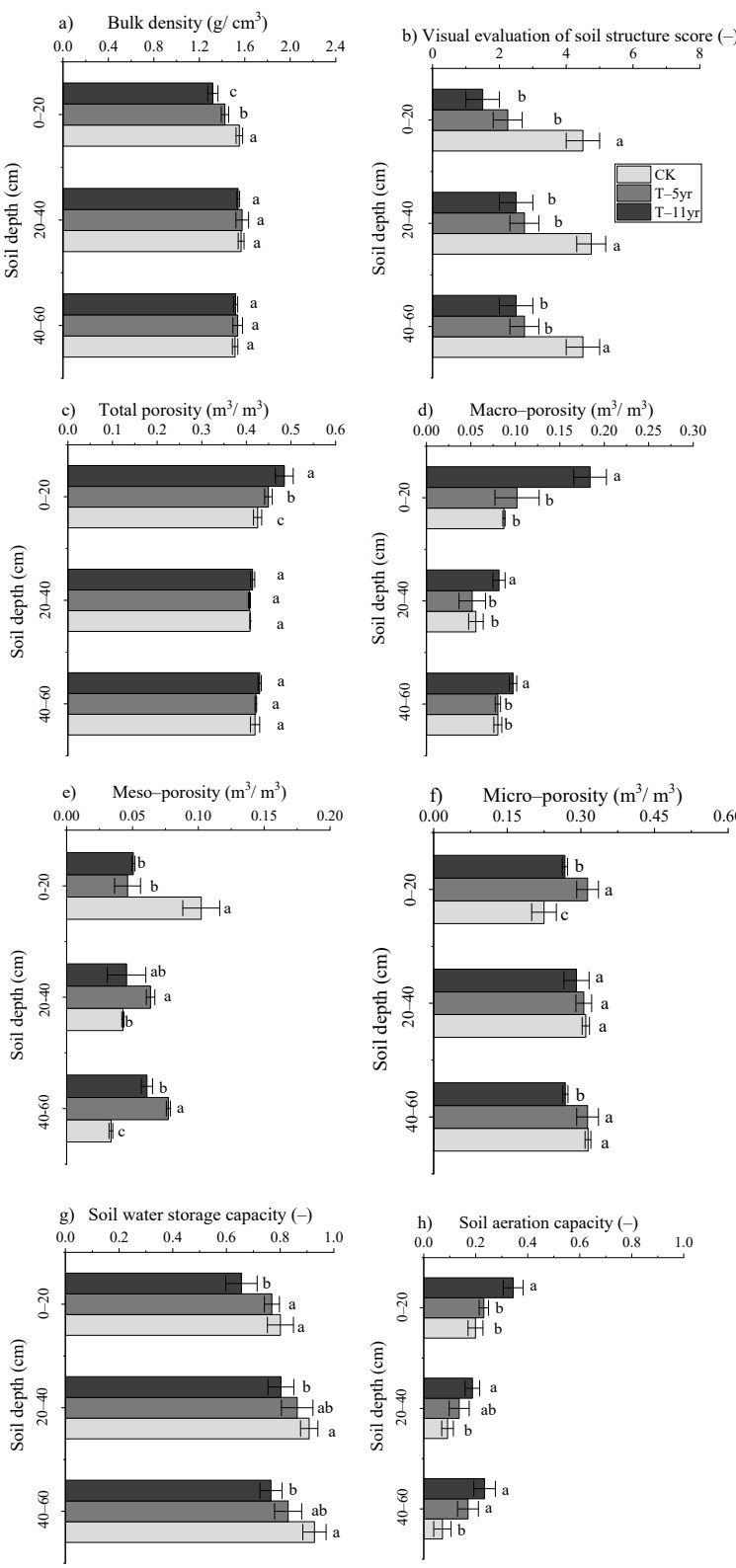

**Figure 2.** Soil bulk density (BD; (**a**)), visual evaluation of soil structure (VESS; (**b**)), total porosity (TP; (**c**)), macro–porosity (MA; (**d**)), meso–porosity (ME; (**e**)), micro–porosity (MI; (**f**)), soil water storage capacity (SWSC; (**g**)), and soil aeration capacity (SAC; (**h**)) for barren land (CK), 5–year–old (T–5yr), and 11–year–old tamarisk plantation land (T–11yr). The bars without the same letter indicate the difference is significant ($p \leq 0.05$). Values represent means ± S.E ($n = 4$).

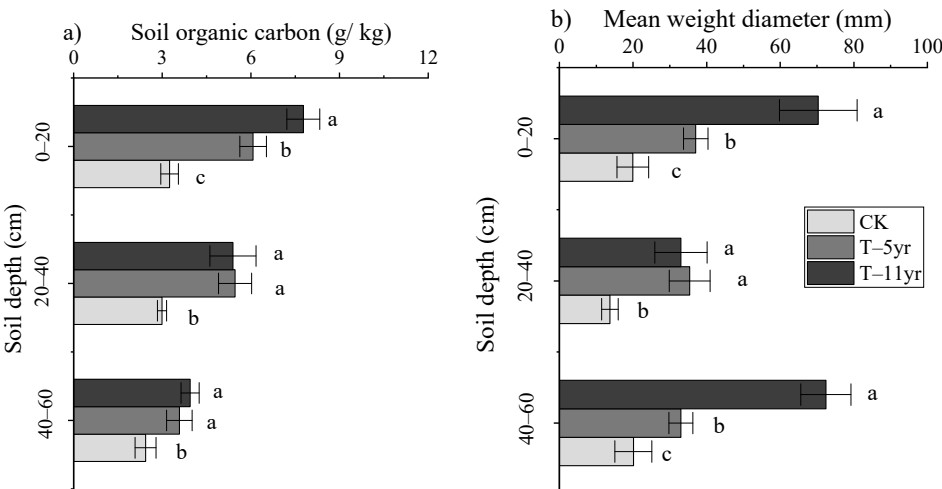

**Figure 3.** Soil organic carbon (SOC; (**a**)) and mean weight diameter (MWD; (**b**)) for barren land (CK), 5–year–old (T–5yr), and 11–year–old tamarisk plantation land (T–11yr). The bars without the same letter indicate the difference is significant ($p \leq 0.05$). Values represent means $\pm$ S.E ($n = 4$).

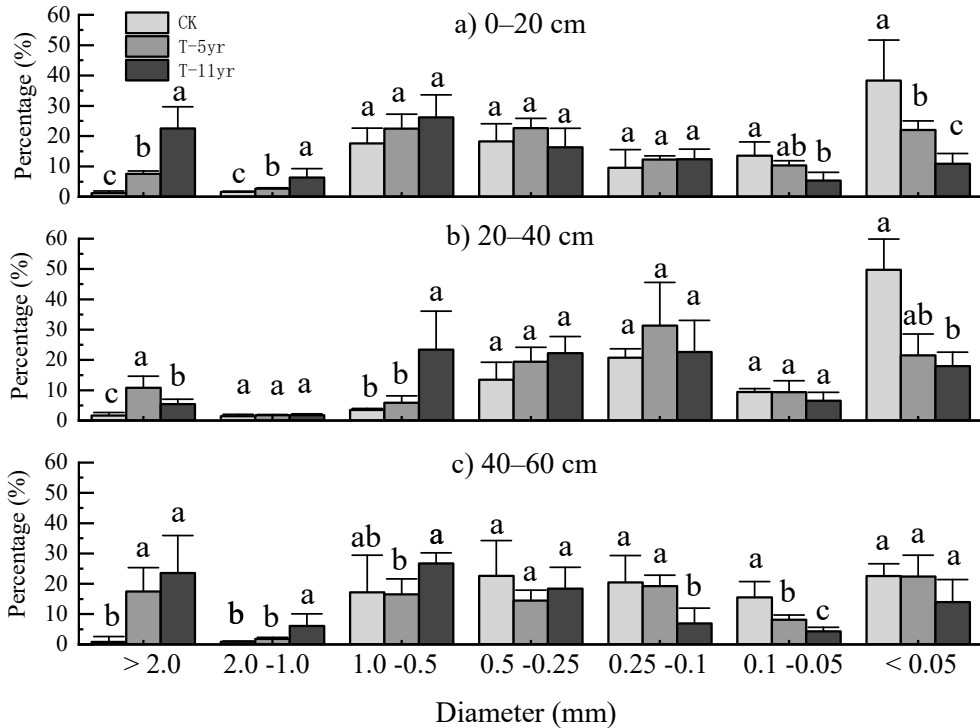

**Figure 4.** Percentage of water-stable aggregate size distributions for barren land (CK), 5–year–old (T–5yr), and 11–year–old tamarisk plantation land (T–11yr). The bars without the same letter indicate the difference is significant ($p \leq 0.05$). Values represent means $\pm$ S.E ($n = 4$).

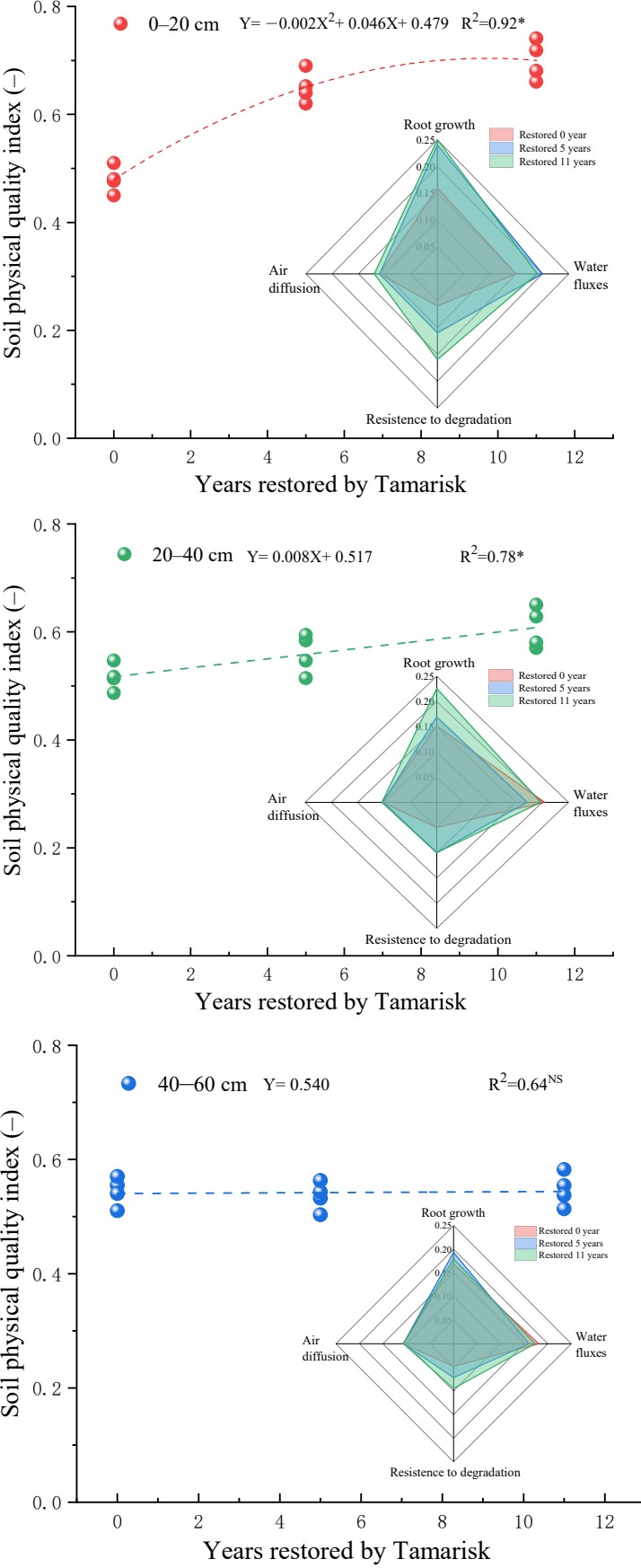

**Figure 5.** Changes in soil physical quality index (SPQI) and soil physical functions with years restored by tamarisk. *, significance level at $p \leq 0.05$; NS, no significance.

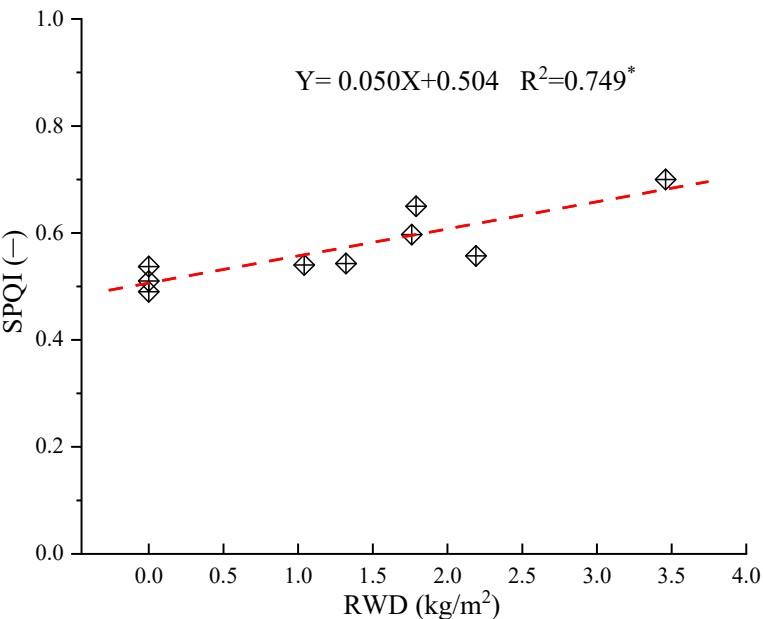

**Figure 6.** Relationship of soil physical quality index (SPQI) with root weight density of vegetation (RWD) in coastal saline land. *, significance level at $p \leq 0.05$.

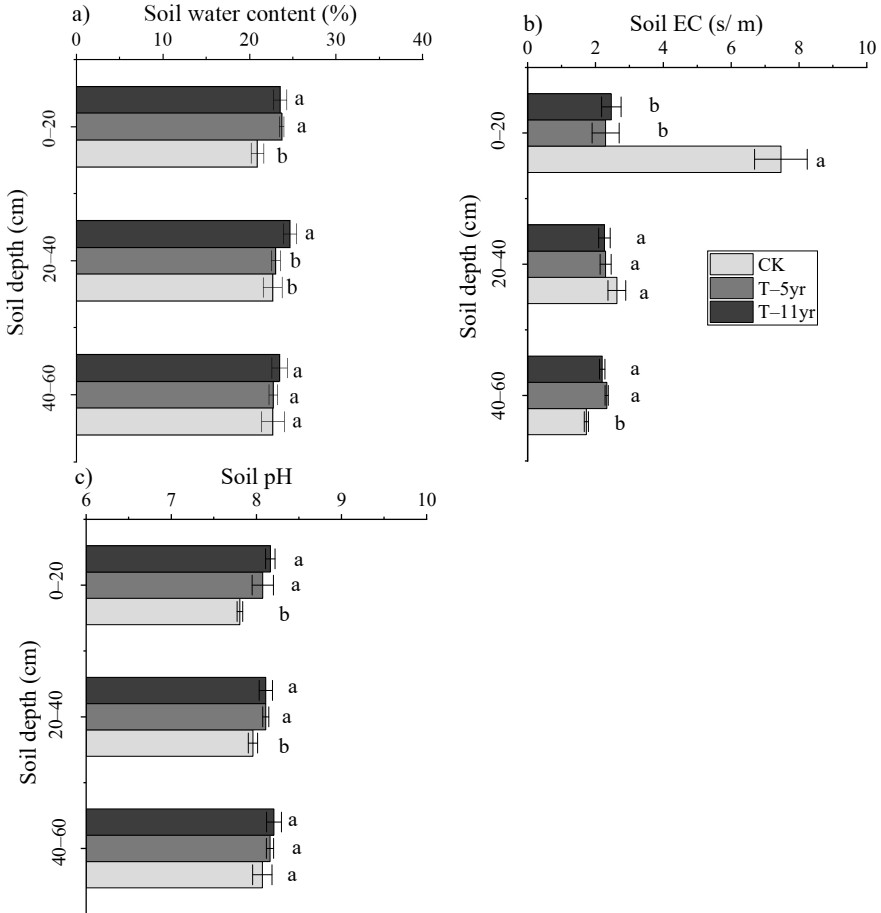

**Figure 7.** Soil water content (SWC; (**a**)), soil EC (**b**), and pH (**c**) for barren land (CK), 5–year–old (T–5yr), and 11–year–old tamarisk plantation land (T–11yr). The bars without the same letter indicate the difference is significant ($p \leq 0.05$). Values represent means $\pm$ S.E ($n = 4$).

**Table 3.** The content of sum of $Na^+$ and $K^+$, $Ca^{2+}$, and $Mg^{2+}$ content for barren land (CK), 5-year-old (T-5yr), and 11-year-old tamarisk plantation land (T-11yr).

| Soil Layer | Sum of $Na^+$ and $K^+$ Content (g/kg) | | | $Ca^{2+}$ Content (g/kg) | | | $Mg^{2+}$ Content (g/kg) | | |
|---|---|---|---|---|---|---|---|---|---|
| | CK | T-5yr | T-11yr | CK | T-5yr | T-11yr | CK | T-5yr | T-11yr |
| 0–20 | 9.39 ± 1.81 a | 3.22 ± 0.51 b | 3.34 ± 0.63 b | 0.30 ± 0.03 a | 0.22 ± 0.08 a | 0.23 ± 0.11 a | 0.55 ± 0.08 a | 0.13 ± 0.05 b | 0.14 ± 0.06 b |
| 20–40 | 4.56 ± 0.56 a | 3.64 ± 0.13 b | 3.66 ± 0.08 b | 0.12 ± 0.02 b | 0.19 ± 0.05 a | 0.12 ± 0.01 b | 0.07 ± 0.01 a | 0.12 ± 0.03 a | 0.07 ± 0.00 a |
| 40–60 | 2.95 ± 0.16 b | 3.70 ± 0.14 a | 3.58 ± 0.02 a | 0.07 ± 0.01 b | 0.16 ± 0.03 a | 0.14 ± 0.05 a | 0.05 ± 0.00 b | 0.10 ± 0.02 a | 0.08 ± 0.02 ab |

Values represent means ± S.E (*n* = 4). Values at each treatment in the same layer followed without the same letter indicate the difference is significant ($p \leq 0.05$).

**Table 4.** The content of $Cl^-$, $SO_4^{2-}$, $HCO_3^-$ for barren land (CK), 5-year-old (T-5yr), and 11-year-old tamarisk plantation land (T-11yr).

| Soil Layer | $Cl^-$ Content (g/kg) | | | $HCO_3^-$ Content (g/kg) | | | $SO_4^{2-}$ Content (g/kg) | | |
|---|---|---|---|---|---|---|---|---|---|
| | CK | T-5yr | T-11yr | CK | T-5yr | T-11yr | CK | T-5yr | T-11yr |
| 0–20 | 14.66 ± 2.36 a | 3.84 ± 0.22 b | 4.02 ± 0.33 b | 0.27 ± 0.03 c | 0.38 ± 0.01 a | 0.35 ± 0.01 b | 2.42 ± 0.14 a | 2.27 ± 0.64 a | 2.35 ± 0.57 a |
| 20–40 | 5.08 ± 0.83 a | 3.90 ± 0.29 b | 3.72 ± 0.14 b | 0.38 ± 0.02 a | 0.38 ± 0.01 a | 0.36 ± 0.01 a | 2.91 ± 0.14 a | 2.9 ± 0.10 a | 2.91 ± 0.20 a |
| 40–60 | 2.56 ± 0.19 b | 4.00 ± 0.30 a | 3.66 ± 0.17 a | 0.40 ± 0.03 a | 0.39 ± 0.01 a | 0.38 ± 0.01 a | 2.58 ± 0.19 a | 2.84 ± 0.08 a | 2.85 ± 0.02 a |

Values represent means ± S.E (*n* = 4). Values at each treatment in the same layer followed without the same letter indicate the difference is significant ($p \leq 0.05$).

### 3.4. Rainfall-Induced Salt Leaching

As shown in Figure 8, the precipitation and groundwater table increased sharply during the June to October (rainy season) in the study site. Intensive rainfall significantly reduced the soil salt content (SAC) in 0–80 cm layer for both T-5yr and T-11yr (Figure 9). In 0–20 cm, the SAC decreased from 13.8 to 2.7 g/kg in T-5yr and 15.0 to 3.5 g/kg in T-11yr by rainfall-induced salt leaching. After the rainy season, the SAC of T-5yr and T-11yr in 0–20 cm was lower than that in deep soil layers. For CK, intensive rainfall had no significant changes of SAC in all layers. The SAC of CK in 0–20 cm was 34.2 and 31.7 g/kg, before and after the rainy season.

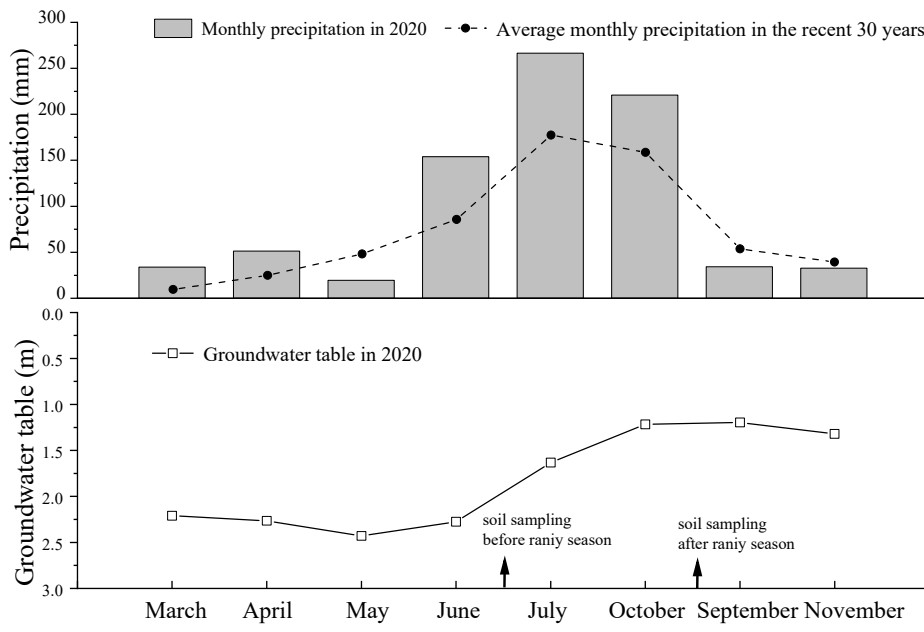

**Figure 8.** The precipitation and groundwater table of study site from March to November in 2020.

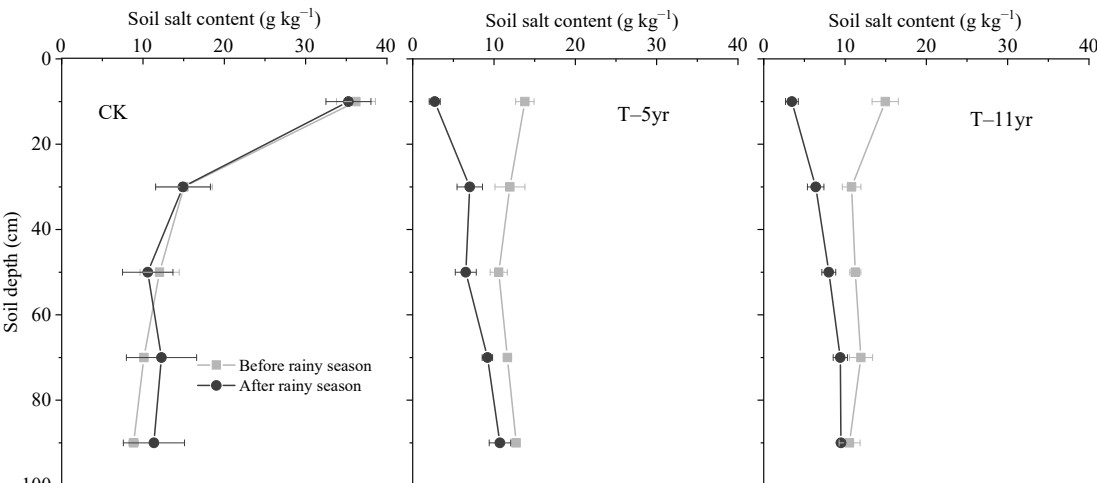

**Figure 9.** Variation of soil salt distribution in profile before and after rainy season for barren land (CK), 5–year–old (T–5yr), and 11–year–old tamarisk plantation land (T–11yr) in 2020.

## 4. Discussion

In coastal regions, elevated salts in soil accelerated the land degradation [42,43]. During salinization, soil $Ca^{2+}$ and $Mg^{2+}$ were replaced by $Na^+$ at cation exchange sites, affecting the expansion and dispersion of soil and resulting in the viscosity texture and poor soil structure [29,44]. In fact, this condition could be improved if vegetation were restored [45,46]. Vegetative bioremediation and phytoremediation have become potential approaches as alternatives to physical, hydro- technology, and chemical amendments, essentially by growing salt-tolerant plant species. In this study, planting tamarisk positively influenced on salt-affected soil in coastal land. Tamarisk significantly increased SOC and thereby promoted the soil water-stable aggregates formation (Figure 3). Owing to the impact of tamarisk on reducing the small soil aggregate percentage (<0.1 mm) and increasing the large soil aggregate percentage (>1.0 mm), soils in tamarisk plantations had higher MWD compared with that in barren land (Figure 4). The changes of soil aggregates resulted in the soil structure improvement, and soils with higher TP and lower BD were considered good for supporting plant growth [41,44]. In agreement with the results of Edrisi et al. [47], vegetation plantation greatly improved the SOC and soil porosity in saline land. SOC and MWD were considered to dominate the soil resistance to degradation [37]. Our study indicated tamarisk restoration enhanced the self-maintenance capacity of degraded soil in coastal system. Moreover, this ecological benefit was time cumulative, because SPQI was positively related with the year of tamarisk restored in 0–20 and 20–40 cm layers (Figure 5). Tamarisk-induced soil physical change was heterogeneous in soil profile, which was determined by the heterogeneity of tamarisk roots distribution (Figure 6). With less root distribution in 40–60 cm, the SPQI of T-5yr and T-11yr was the same as that of CK (Figure 5).

$Na^+$ and $Cl^-$ are monovalent charge ions which migrate with water easily [11]. Thus, soil physical properties associated with water flux are important in regulating soil salinity [48]. In this study, following tamarisk restoration, the topsoil MA was increased and ME (related to soil water store capacity) was decreased. It could promote water infiltration and the removability of soil salts. After the rainy season in 2020, obvious salt-leaching and the decrease of soil salt content was observed in tamarisk plantations (Figure 8), while in barren land (CK), intensive rainfall had little effect on soil salinity because of the low score (0.15) of soil physical function for water flux. In addition, the increase of soil pH after planting tamarisk was suggested indirectly related with the recurring salt-leaching process.

Soil restoration and desalinization were the two main objects for vegetative bioremediation in salt-affected land. The effect of tamarisk vegetation on soil physical improvement was in agreement with other woody species, such as *Albizia lebbeck*, *Casuarina equisetifolia*,

*Cordia dichotoma*, etc in the saline land of western India [47]. Artificial plantation greatly reduced saline soil BD and improved the soil porosity by root-soil interactions, which greatly determined the saline soil reclamation level [47]. Seenivasan et al. [49] reported that soil EC gradually reduced after *Eucalyptus* restoration in saline land. However, the consistent decrease of soil salt content in tamarisk plantation was not observed in this study. After 11-years restored by tamarisk, topsoil EC stabilized at 2.2 s/m, which is similar to that of 5 years after restoration (Figure 6). We suggested that it was because of the study site specific geography (shallow saline water stable and dry climate) and the characteristics of tamarisk species (salts absorption and excretion). It was reported that tamarisk showed strong tolerance to saline conditions, which could be explained by the salt glands in leaves, and this species could absorb the soil salts along with water-transport in organism and simultaneously avoid salt toxicity [50]. The increase in soil salinity was accompanied with the increased accumulation $Na^+$ and $Cl^-$ within the tamarisk. These ions also could serve as cheap osmotica to improve the plant-water relations [51,52]. Therefore, in this study, tamarisk reduced large amounts of topsoil ions content, especially for $Na^+$ and $Cl^-$, by root absorption (Tables 3 and 4). Moreover, combined with the soil physical improvement on salt-leaching in tamarisk plantation, the soil desalinization in topsoil was of significance (Figure 7b). However, it should be noted that soil salt distribution was tightly related with the local climate. In the arid or semi-arid regions, tamarisk may lead to the water loss and soil salts enrichment in surface soil [53]. We suggested that without washing and drainage effects of rainwater, the salt recycling effect of tamarisk was limited because the withered branches and leaves would deliver back the enriched salts to soil.

In barren land (CK), salt accumulation of topsoil was observed (particularly higher soil EC in topsoil than that in deep layers), while in tamarisk plantation, there was homogeneity of salt distribution in soil profile. As we hypothesized, if soil physical obstruct was removed, the soil salts distribution will change naturally and then develop into a new balance in coastal ecosystem for a long time-scale. In the prophase of tamarisk establishment, coastal soil was rapidly restored and gradually desalinized due to soil physical improvement. Moreover, because of the rainfall induced salt-leaching in tamarisk plantations, the soil salt content of topsoil was reduced which offset the potential risk of naturally salt accumulation in study site.

Soil variables are an important factor in shaping the structure and composition of communities at a regional scale in addition to climate and plant characteristics [54]. Although after tamarisk restoration the topsoil salt content (3–15 g/kg) was still too high for the most crops growth, the improved soil made it suitable for halophytic cultivation, the esculent plants *Mesembryanthemum crystallinum* [55], *Sueada salsa* [33], *Atriplex triangularis* [56] for food production, or the halophyte production with fuel, feed, and fiber value [57]. Moreover, planting *Cistanche deserticola* in tamarisk forest was another new medicinal crop industry with high economic performance. In addition, as shelter forest, the tamarisk plantation could protect against the sandstorms and storm surge for the safe living environment in coastal areas. The improvement of plant community compositions and the soil resistance to degradation would also help for the stability of coastal ecosystem, especially during the potential extreme climate changes. In summary, this study indicated the feasibility and security of planting tamarisk in Bohai coastal saline land and other similar habitats of the world for ecological restoration. Meanwhile, this study explored a new regulation mechanism across vegetation–soil–environment in coastal ecosystems.

## 5. Conclusions

Plantation of tamarisk positively influenced the soil environment in coastal saline land. It increased soil oragnic carbon, promoted soil large aggregate formation, and reduced soil bulk density and soil structure score. Therefore, the physical functions of topsoil were enhanced by tamarisk, especially in the aspects of supporting root growth and the resistance to degradation of soil. The effect of tamarisk on soil large pore structure contributed to the decrease of topsoil salinity and promoted the homogeneous salt distribution. Tamarisk

plantation land benefited from the enhanced soil resistance to salt accumulation, and there was little potential risk of salinization. The finding of this study demonstrated a promising potential of tamarisk plants by altering soil properties in favor of our goal to change salt-affected coastal land into a fertile field and a viable coastal ecosystem.

**Author Contributions:** Conceptualization, K.G.; methodology, K.G. and C.Y.; software, J.L.; validation, T.H.; formal analysis, K.G. and C.Y.; investigation, J.L.; resources, J.L.; data curation, X.F.; writing—original draft preparation, J.L. and X.F.; writing—review and editing, K.G. and X.L.; visualization, T.H.; supervision, C.Y.; project administration, K.G.; funding acquisition, K.G. and X.L. All authors have read and agreed to the published version of the manuscript.

**Funding:** This research was funded by the National Key Research and Development Program of China (2021YFD1900904) and the key research and development program of Hebei province (20327002D, 21326411D).

**Institutional Review Board Statement:** Not applicable.

**Informed Consent Statement:** Not applicable.

**Data Availability Statement:** Not applicable.

**Conflicts of Interest:** The authors declare no conflict of interest.

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
