# Peer review of "Long-Term Effect of Tamarisk Plantation on Soil Physical Properties and Soil Salt Distribution in Coastal Saline Land"

_agronomy, doi:10.3390/agronomy12081947_

Round 1

Reviewer 1 Report

Your manuscript was revised according to my comments. However, you did not compare the capacity of desalination of other woody species in the discussion. On the research of desalination of woody species, there were several cases using Eucalyptus (e.g. Seenivasan et al. 2015, Environmental Geochemistry and Health). Moreover, there was a resent study conducted using several woody species of desalination (Edrisi et al. 2021, Land Degradation and Development). This journal is international, and you have to estimate your results by comparison of similar researches of other countries.

 Moreover, you could not abbreviate the calculation of VESS value by quotation because readers are interested in the method of calculation. You must show the method in your manuscript.

Author Response

Thanks for the comments, and we have improved the manuscript as follows,

1, As your comments, it is essential to compare the soil changes performance between tamarisk and other woody species in saline land. Thus, we improved the discussion about this part in Line 337-339, 363-371.

2, We have added the brief description of the calculation method for VESS value, in Line 156-161.

We have tried our best to improve the manuscript with changes noted.

Thanks again! With best wishes!

Reviewer 2 Report

I checked out carefully through the manuscript's revised version and the authors' responses to each comment. I believe the revised MS has been significantly improved and the authors have seriously focused on improving the recommendations. However, before publication, the MS should be fully revised by a native English speaker. The authors have mentioned terms like as obviously, fortunately, almost, durative, etc. which do not reflect a scientific approach.

Author Response

Thanks for the comments, and your kind work! The language of this manuscript has been improved with the aids of native English speakers, which we hope it meets the requirements of the Journal. With best wishes!

Reviewer 3 Report

The revised version of the manuscript submitted by Li et al. it is much improved compared to the previous version. In particular, are appreciated the following issues:

- integration of the environmental information relating to the experimental site;

- some datils on the vegetation occurring on the unmodified site,

- rectification of the definition "saltmarshes", and

- the new photos suitable to visualize the positions of the experimental areas compared:

- new comments concerning the soil salinity profile in connection with the rising of the salty groundwater and evapotranspiration rate.

These changes allow a better understanding of the environmental conditions in which the experimental project was carried out.

Furthermore, various changes to the text, especially in the introduction, have improved the quality of the contribution and made it more understandable.

There remain some doubts which, in my opinion, it would be good to resolve before recommending its publication.

1. What is the ultimate purpose of tamarisk planting?

Following the corrections introduced, it is clear that the original environment is a salinized habitat vegetated by halophytes. The salt content of around 30 g/kg is very high and it is not clear if these area can be classified as wetlands. Among the plants originally present, Phragmites australis requires wet soils, whereas it does not survive salinities so high. On the contrary, Suaeda spp. are very halotolerant. However, these are clearly natural saline environments, not degraded by environmental changes due to anthropization or industrial activities, so the purpose of the interventions remains a bit unclear. The work results show that tamarisks allowed to reduce the salinity and modified the structure of the soil (which is the object of the study), but not enough to make the soil arable. In particular, the maximal reduction of salinity is already achieved within 5 years following planting. This salinity value fluctuates between 3 and 15 g/kg depending on the rainfall. These values ​​are not suitable for common agriculture purposes. Since the salinization of the soil occurs due to the rising of the salt water, it seems that the topsoil reaches a state of equilibrium between the rising of the underground salt water and the tamarisk salt absorption. Furthermore, removing the tamarisk trees to attempt agricultural plant crops would progressively return to the initial salinity. To conclude, the soil improvement achievement has no other utility than to substitute a natural saline soil with an artificial tamarisk plantation. The latter could prevent soil erosion or other environmental problems, but the data collected in the control area do not show any problem in this sense.

In other words, the proposed contribution would have a much greater impact if this aspect were better discussed, which finally defines the usefulness of the use of tamarisk. For example: would it be possible to use tamarisks to make the environment cultivable with halophytes suitable for forage? In the literature, halophytes that can be cultivated for this purpose are described which adapt to salinity between 5 and 20 g/kg with good producions. The harvest of these halophytes, with their salt content, should guarantee the maintenance of the saline balance achieved through the initial use of tamarisk. Furthermore, recent paper pointed out that saline stress increases the synthesis of active compounds (carotenoids, polyphenols, etc.) by halophytes, and these are useful for improving the human nutrition. 

Apart from this idea, proposed for discussion purposes only, it would be important to give greater importance to the aims of the adopted  environmental technique in relation to the soil quality data that are discussed.

2. Improve the discussion relating to the data of fig. 2

In my previous revision, I observed that there is an apparent inconsistency between some data showed in fig. 2, but maybe I made my comment unclear and I apologize. So, I try to explain myself better. Please focus on the data relating to the 0-20 cm layer, which seems to me the most interesting.

Fig. 2 shows that T-11yr has an increased total porosity compared to CK, mainly due to the increase in macroporosity, and to a lesser extent due to the microporosity, while the mesoporosity decreased significantly. This can be consistent with the overall loss of bulk density. However, these soil structure changes seem inconsistent with the fact that T-11yr and CK maintain the same "soil water storage capacity" and "soil aeration capacity" (0-20 cm). At least one of these two soil traits (or both) should be significantly different between T-11yr and CK, in my opinion, as effect of the changes in porosity. I think it would be appropriate to propose a reasoned interpretation, also in light of the fact that T-5yr, shows porosity data that do not seem consistent with a linear progression of the effects due to the tamarisk.

3. Methods: statistical analysis description

It is recommended to report also the statistical tests used to verify the assumptions, i.e. data normal distribution and homoskedasticity. 

I enclose a MS-pdf with notes for further support to the authors

Author Response

Thanks very much for the comments and your careful work which is of importance to improve our manuscript.

(1) The study site was a salinized dry land and could not be classified as wetlands. As your comment, after tamarisk restored, the soil salt content in coastal land was still too high for field crops. Thus, the ultimate purpose of tamarisk planting was to, a) protect the living environment against the sandstorms and storm surge in coastal areas, b) improve the soil quality for halophytic cultivation, in aims of food, fuel, feed, or fibre production. c) enhance the land resistance to the potential degradation, which was mentioned in the manuscript Line 409-424.

(2) In fact, we mostly appreciated for this comment which help us to correct the mistake in the manuscript. According to your detailed explain, we also felt confused about the inconsistent changes of SWSC and SAC after planting tamarisk (Fig.2 g; h). Thus, after a check for the original data, we found there was a mistake during the calculation of SWSC/ SAC. Then, we have corrected the Fig.2 (Line 256-260; 682-683). Over restored timing, the SWSC was gradually reduced after planting tamarisk and the SWSC of T-11yr was significantly lower than that of CK in all soil layers.

(3) We have added the statements about the statistical tests on the assumptions of data normality and homogeneity of the variances of the residuals in Line 229-231.

The other mistakes in manuscript were also corrected, according to the comments.

Thanks again! With best wishes!

Reviewer 4 Report

This is very important and extensive research with significant results.

Generally, English should be corrected in order to make the paper readable.

Major point: The authors covered some very important topics but did not widely discuss salt recycling which is the major disservice here. Neither did they mention Tamarix's mechanism of salt management inside of its plant parts, which is very significant in explaining the salt reduction, especially in topsoil parts. 

Please address some minor points given below.

Line 96: survive not survival 

Line 125: correct to: which is tightly 

Correct Tamarix plantation lands to Tamarix experimental plots

Line 218: small caps for Bulk

Correct the citation Roberta Q et al (2020)

This sentence is unfinished, generally, the text is difficult to follow due to poor English. Please add the bolded part:

Tamarisk (T-5yr and T-11yr) significantly reduced the soil BD in 0-20 cm, where the following rank was observed T-11yr (1.32 g/cm3 )< T-5yr (1.43 g/cm3 )< CK (1.56 g/cm3).

line 286 scores was  were

Line 303: While In 0-20 cm

The same line, correct increased.

Line 352: correct reduced

Conclusion section: 

Benefited from the enhanced soil resistance to salt accumulation, there was little potential risk of salinization in tamarisk plantation land at the study site. The finding of this study demonstrated a promising potential of tamarisk plants by altering soil properties in the favor of our goal to make as a fertile field and a viable coastal ecosystem.

Please discuss these points prior to the conclusion section, referring to Tamarix's mechanism of salt management inside of its plant parts, which is very significant in explaining salt reduction, especially in topsoil parts. 

Author Response

Thanks for the comments. We have corrected the mistakes and improved the language in new manuscript with changes noted. The tamarix's mechanism of salt management also have been added in discussion (Line 374-387) for a better explain about salt reduction in topsoil, as recommended.

Thanks again! With best wishes!

This manuscript is a resubmission of an earlier submission. The following is a list of the peer review reports and author responses from that submission.

Round 1

Reviewer 1 Report

The manuscript proposed by Li et al. aims to study the physical-chemical changes of salinized soils occurring following planting with tamarisk, a halophyte able to tolerate the soil salt absorbed with the water and to secrete it through the leaves.

The physical properties of the studied soils and the distribution of some mineral salts are analyzed and discussed in relation to the presence of plants and the density of their roots up to a depth of 60 cm.

Title: "Long-term effect of tamarisk on soil physical properties and soil salt distribution in coastal ecosystem"

The title should concisely but clearly summarize the subject of study. This title refers mentions the effect of the tamarisk presence on the soil physical properties in a generic coastal ecosystem. Coastal zones include many and different ecosystems of a great ecological value, while in this case the study is focused on the problem of recovering salinized soils. Furthermore, the authors use the term “coastal ecosystem” in the title, but they do not provide any ecological detail in the following description of the sites of interest. This imprecise, sometimes improper, language is quite frequent throughout the manuscript and represents an important weakness which, in some points, compromises the reliability of the proposed statements.

Introduction

The introduction presents the problem under study in an excessively summary way. The environmental context in which the activities were performed are not described in a sufficiently detailed way, referring to coastal environments without details useful to better understand the proposed case study.

The plant used for this study is mentioned as “tamarisk” without ever identifying it with the scientific name, assuming that the species ordinarily present in the area is the only one identifiable with the common name of tamarisk. It should be considered that each species has its own biology, which is relevant for the proposed goals. The concept of "fertile islands effect" is used (line 54) without clarifying its meaning, forcing the reader to check the cited bibliographic references to find an explanatory definition.

Line 66-67: this statement implies that the salt rises from the water table and therefore it would be logical to expect the salt gradient to decrease towards the surface of the soil, where it is washed away by rain. However, the vertical ‘Na and K’ distribution showed in TAB. 3 for the control area (CK) has an opposite profile. This paradox creates is not discussed and this jeopardize the interpretation of all data subsequently proposed.

Experimental design

The study area is defined as a coastal salt marsh (line 83) without any further detail. Salt marshes are natural environments of great ecological importance and threatened by anthropogenic activities all over the world. For this reason, they are subject to protection according to the Convention on Wetlands of International Importance, Waterfowl Habitat (Ramsar Convention), the Convention on Biological Diversity, and Agenda 21 adopted by the 1992 United Nations Conference on Environment and Development. Furthermore, contrary to what is reported in the introduction, salt marshes are considered among the environments with the highest productivity among those known (Vadas, R. L., Garwood, P., & Wright, W. (1988). Gradients in salt marsh productivity in the Wells National Estuarine Research Reserve. NOAA Technical Memorandum n. 20, 87 pp.).

From the brief description, however, it seems that the study sites were not salt marshes but arid coastal areas affected by a process of salt accumulation. This is just a hypothesis, as the historical-ecological description of the environmental context is completely missing, nor is it clarified how far the site are from the coastline, or how much they are elevated over the maximum tidal height (syzygal tide). In FIG. 1b, it is shown that CK has a vegetation cover, apparently not less than 50%. It would be appropriate to provide a brief description of the plants occurring in this environment.

The surfaces of the study sites are not quantified and there is no map showing how they are positioned relative to each other. Lacking soil sampling in T-5yr and T-11yr prior to tamarisk planting, it is important to demonstrate that CK is indeed representative of the soil condition prior to the interventions. This obviously depends on how close they are to the tamarisk-planted sites.

Results and discussion.

Statistical analyzes were limited to the minimum threshold value of significance (p ≤ 0.05), labelling the results with “**” instead of the conventional symbol “*”. On the other hand, it would also be useful to highlight when the difference in the measured values ​​is very significant (p ≤ 0.01), which in fact is generally indicated with the symbol “**”.

In general, the discussion appears to be insufficient in some points. As has already been highlighted above, the vertical salt distribution (Na and K; Cl) showed in Tab. 3 for CK appears to be not very consistent with the rising of salt from the water table. In addition to this, in the 0-20 layer, the salt concentration at T-5yr is equal to T-11yr: authors highlight in FIG. 7 that the salt level decreases in the first 5 years and then tends to rise again. Since the salt phyto-extraction effect continued in T-11yr for more than twice as long as in T5yr, this conclusion seems illogical, or at least it should be commented in more detail. However, this conclusion is incorrect in my opinion, as the variation of salt concentration between T-5yr and T-11yr is only apparent, as there is no statistical difference between the data of these two sites. Indeed, the small increase of salt detected at T-11yr must be considered as due to a sampling effect.

In FIG. 2, it is observed that the soil total porosity (TP) was increased from 0.42 m3/ m3 (CK) to 0.45-0.49 179 m3/ m3 (T-5yr and T-11yr) in the 0-20 cm layer (Fig.2c). Even considering the distribution of meso- and micro-porosity, it appears difficult to explain the results reported for the soil water storage (SWSC) capacity and soil aeration capacity (SAC) (see lines 184-186). In fact, in the 0-20 cm layer, which is the layer most affected by the tamarisk presence, SWSC and SAC of T-11yr are the same of CK, while they are both different from T-5yr, and this seems inconsistent with the other soil properties detected.

References

This section needs to be revised to comply the editorial guidelines.

General comment

The work presents a significant dataset demonstrating that the tamarisk planting can significantly affect the soil physical properties of salinized soils. However, the proposed data analysis has some important shortcomings, whereas some inferences are not justified or even in partial contrast with the reported data. The general conclusion that tamarisks produced a decrease in the salt content and modified the soil physical properties are objectively true, but some issues concerning the data interpretation appear unresolved. An analysis of the soils prior to the planting of the tamarisk would have been essential and perhaps would have partially modified the interpretation of these results.

It should also be noted that the preliminary ecological description of the sites is almost absent. Based on my understanding, the sites cannot be described as salt marshes. Furthermore, some non-standard concepts such as "salt islands effect" and "fertile islands effect" are introduced without appropriate preliminary definition, and then used to discuss the results with arguments susceptible to be criticized, on the basis of a different interpretation of the concepts mentioned.

I enclose a MS-pdf with notes for further support to the authors

Reviewer 2 Report

This manuscript tries to evaluate changes in soil physical properties in a salt marshes restoration initiative. Based on chemical attributes, soil physical quality index, soil salt distribution, and salt leaching the authors observed changes in soil physical properties in tamarisk plantations of different ages and compared with barren land as control. Their results clarify that tamarisk plantations are capable to increase soil quality which may be applied for restoration initiatives. I propose that this manuscript should be published after a major revision.

General comments:

My main concerns are regarding the absence of a natural area for comparison, the absence of historical uses of these areas, and the physical indicators applied. The physical indicators that were used in the quality index are associated with water fluxes and the storage capacity of water. However, the study site is a salt marsh subjected to constant flooding. I wonder if these physical indications are representative since fluxes and water may be altered daily. The absence of a natural area for comparison, calls into question the scores ranging from 0 to 1 in the SPQI. What value is used as a reference to be considered 1, for example? Additionally, the discussions could be improved using previous studies to discuss reactions, mechanisms, and processes.

Specific comments:

Please see the manuscript files with specific comments.

Reviewer 3 Report

Although your research had a weak point on the absence of site replication, your results were novel, and had a value to publish in Agronomy. I think your manuscript should publish after minor revisions as following.

1.    I think that you should compare your results with previous literatures. I understood that tamarisk plantation had high capacities of soil improvement, salt desalination, and increase of organic substances. However, I could not estimate these capacities of tamarisk. You should survey several literatures on the capacity of desalination of other woody species in the discussion. I expect that you can explain the superiority of tamarisk.

2.    You have to show the method of the calculation of VESS value.

3.    On the calculation of soil physical quality index, you used four factors (f (i-iv) ). However, I could not confirm the original data. You should add these data in the results.

4.    The abbreviation of “the stratification rate of soil salinity” was not unified (SRSA and SRSS). You have to unify the description of abbreviation.

5.    On the explanation of Fig. 7, you show the data of SRSS for the CK treatment.

6.    It is better to show the data of seasonal precipitation at research site because I could not imagine the results of 3.4.

7.    I could not understand the reason why contents of Na and K were combined. You should show the contents of Na and K, respectively.

8.    I could not understand the reason why you could show root weight density for the CK treatment. You should explain the method.

9.    I predicted that you used the multiple statistical analysis for the greater part of figures and tables (presence of alphabet). However, you did not explain in the Materials and Method. You must show what kind of statistical analysis is used.

10. You have to show the number of individuals in each figure.

11. In Fig. 5, radar chart of 40-60cm was used the different word (plant growth). You have to unify to “root growth”.

12. On the horizontal scale of soil pH (Fig. 6c), you should change the value of scale (e.g. 7-9).

13. On the explanation of Fig. 9, you have to write in the results. Also, you have to show the method of analysis in the Materials and Methods.

14. In your discussion, the contents of introduction were included (e.g. L. 238-244, 258-260). Your discussion should be centered around your results.

15. On the explanation of “fertile islands” (L. 276-277) and “salt islands” (L. 283-285). You should explain using your results.